# Nitrogen uptake preference of cotton (*Gossypium hirsutum* L.)

**James O. Latimer[1,2]\*, Mark Farrell[3]\*, Ben C. T. Macdonald[2]**

**1** The Fenner School of Environment and Society, The Australian National University, Canberra, Australian Capital Territory, Australia, **2** Agriculture & Food, Commonwealth Scientific and Industrial Research Organisation, Black Mountain, Canberra, Australian Capital Territory, Australia, **3** Agriculture & Food, Commonwealth Scientific and Industrial Research Organisation, Waite Campus, Urrbrae, South Australia, Australia

\* James.Latimer@anu.edu.au (JOL); mark.farrell@csiro.au (MF)

## Abstract

Low molecular weight (LMW) soil organic nitrogen (N) can be a significant source of N in commercial cotton (*Gossypium hirsutum* L.) systems, potentially comprising a meaningful portion of N uptake in Australian irrigated cotton. Cotton obtains the majority of its N from the soil N pool rather than directly from fertiliser—N. Organic N is the major component of the soil N pool. The purpose of this study was to test the ability of *G. hirstutum* to take up different organic and inorganic N forms using isotopically labelled compounds. This was done in a sand matrix to reduce potential for microbial competition and enable a clearer view of the physiological capacity of the plant to access different N forms. The experiment showed that cotton took up inorganic N ($NO_3^-$ and $NH_4^+$) and organic N (alanine and urea) concurrently, with a slight preference towards inorganic N overall. The uptake mechanism for organic carbon (C) associated with the organic N was also examined, showing that alanine—C was taken up linearly, with a consistent internal $^{13}C:^{15}N$ ratio suggesting that some alanine was absorbed intact without extracellular deamination. Overall, the experiment demonstrated that *G. hirstutum* can rapidly and concurrently access different soil N pools, with a slight preference for inorganic N. The uptake mechanisms for organic N and C are complex, differing between compound types, and warrant further investigation. This study expands the list of plants known to utilise organic N to include commercial cotton, with implications for the management of N fertiliser in cotton growing systems.

## Introduction

The cotton fibre produced from *Gossypium hirsutum L.* farming systems is globally the most widely utilised natural fibre [1]. It was estimated by Heffer and Prud'homme [2] in 2014 that the global fibre crop, which is mostly cotton, uses 4.3% of the global

**Data availability statement:** All relevant data will be made publicly within the CSIRO Data Access Portal (https://data.csiro.au). A unique DOI will be generated upon acceptance to be included with the article at the typesetting stage. There will be no restrctions to the data.

**Funding:** The authors gratefully acknowledge the Cotton Research and Development Corporation (https://crdc.com.au/; Grant number CSP1904 awarded to BM and MF) for funding. The funders played no role in the study design, data collection and analysis, decision to publish, or preparation of the manuscript.

**Competing interests:** The authors have declared that no competing interests exist.

production of nitrogen (N) fertiliser (110.4 Tg N) to produce 25.9 Tg of lint [3]. The global N use efficiency index for cotton is 6 kg lint $kg^{-1}$ applied N, well below the recommend optimum range of 13–18 kg lint $kg^{-1}$ applied N [4,5]. This indicates that the global crop is potentially receiving more fertiliser N than required, and may be subsequently impacting environmental processes across the globe [6]. This apparent over fertilisation results in N run-off losses [7], deep drainage [8] and ground water pollution [9], and greenhouse gas production [10]. Brackin et al. [11] showed that despite the significant amounts of applied N fertiliser, cotton crop fertiliser recovery is only between 27–38%. This indicates that a substantial proportion of crop N is derived from the antecedent soil N pool. It was assumed by Brackin et al. [11] that cotton N uptake is from the inorganic pool after mineralisation despite the presence of organic N pool within most soils.

Soil organic N is derived from soil organic matter (SOM), which typically constitutes the largest soil N pool, and contains more C than the atmosphere and global vegetation combined [12,13]. Globally, soils have lost approximately half of their C stocks since the adoption of soil cultivation techniques [14] and therefore also potentially 50% of their N associated with the C in SOM. Soil organic matter is frequently undervalued when considering crop nutrition, with the focus often directed at inorganic N ($NO_3^-$ and $NH_4^+$) sources from fertiliser. To simplify N management, N-bearing molecules are commonly aggregated into different pools based on context-specific functionality, which typically are: inorganic N, dissolved organic N (DON), synthetic N, and insoluble organic N pools. The size of these pools can vary considerably throughout the year [15] because N continually fluxes between these different compounds, with no molecule providing permanent immobilisation [12].

Since the mid-19th century and the dismissal of the humus theory of plant nutrition, the dominant agricultural paradigm has held that plants only take up N from the soil in the inorganic forms of $NH_4^+$ and $NO_3^-$. This limited view of plant nutrition pervaded to the end of the 20th century, likely due to the ease with which inorganic N can be measured and the fact that it is definitely plant-available. An increased understanding of the role microbes play in soil N mineralisation combined with an over-extrapolation of plants' reliance on this process likely also contributed to this outdated concept [16]. In addition, the rise of Haber-Bosch-derived fertilisers at the beginning of the 20th century shifted the focus away from organic soil amendments and fixation as the primary means of increasing plant-available N. With productivity gains so easily achieved by synthetic fertilisers, the research community and industry shifted its focus towards $NH_4^+$ and $NO_3^-$ and maximising yield.

Research demonstrating the importance of organic N in plant nutrition is not new [17–19]. However, it was not until the discoveries that organic soil N can be the principal N pool for some plants [20], and that low molecular weight (LMW) DON molecules can be taken up intact by plants [21,22] that the mainstream view of terrestrial N dynamics shifted to include direct plant uptake of DON from the soil. The uptake of DON has been documented in many plant species, often in concert with inorganic N [23], but has not yet been observed in commercial cotton (*G. hirsutum*). No previous experiments have demonstrated *G. hirsutum*'s ability to take up organic N, nor

its preference for one N species over another. To address this knowledge gap, this paper reports on four experimental hypotheses. The first was that *G. hirsutum* would not exhibit a N uptake preference between the different N species ($NO_3^-$, $NH_4^+$, urea and alanine). The second hypothesis was that alanine—N and alanine—C uptake would correlate, indicating that alanine is taken up whole by *G. hirsutum*. The third hypothesis was that urea—N and urea—C uptake would correlate, indicating that urea is taken up whole by *G. hirsutum*. The fourth experimental hypothesis was that the three *G. hirsutum* varieties (transgenic, non-GM, original landrace) would not exhibit different N uptake behaviours. This experiment was not designed to emulate natural conditions, but rather to assess the plant's physiological capabilities and preferences. Alanine was chosen as a model non-synthetic organic N compound of intermediate turnover time [24], whereas urea is a major form of synthetic N used as fertiliser in cotton growing systems.

## Materials and methods

### Experimental design

Three varieties of *G. hirsutum* were grown to a two- to four-leaf stage in a randomised block design with 10 replicates. The three varieties used were: Sicot 746B3F, a GM current commercial cultivar accounting for more than half (54%) of the 2017−18 Australian summer plantings; Sicala V2, an obsolete non-GM commercial cultivar released in 1994; and Tx III, a Guatemalan landrace accession that represents the native origins of the commercial *G. hirsutum* varieties. Plants were grown in 300 mm deep low-density polyethylene (LDPE) pots, consistent with the rhizotube design used by Hill and Jones [25]. Tubes were open-bottomed, allowing drainage and preventing waterlogging. Pasteurised and washed sand of <1 mm particle size was used as the growing medium to enable control over plant-available N pools and to limit the potential of microbial competition for the applied N compounds. Plants were watered using a nutrient solution mixture that constituted 100% of the plants' soil-derived nutrient supply (Table 1). The plants were grown in glasshouses at the CSIRO Black Mountain site in Canberra, ACT, Australia (35 °16'S 149 °7'E) in January (summer) 2018. The high solar radiation experienced by Canberra in summer required the use of 70% shade cloth, as sand temperatures exceeded 60°C when exposed to direct sunlight. In total, 450 seedlings were analysed, comprising 10 replicates each of 45 unique treatments following the original randomised block design: five isotopically-labelled N and control treatments, three *G. hirsutum* varieties, and

**Table 1. Nutrient solution prepared based on the Hoaglands solution and recommendations from Oliver Knox, University of New England. Mass and mol per cent are calculated excluding oxygen and hydrogen. Solution was pH balanced to 7±0.05 and EC kept below 1000 µS cm-1 to reduce the chance of acidification or salinisation.**

| Element | Mass Per Cent | Mol Per Cent | Compounds |
|---|---|---|---|
| N | 25.17% | 44.29% | $KNO_3$, $Ca(NO_3)_2$, FeNaEDTA |
| P | 3.61% | 2.88% | $KH_2PO_4$ |
| K | 27.38% | 17.25% | $KNO_3$ |
| Ca | 23.38% | 14.38% | $Ca(NO_3)_2$ |
| C | 2.80% | 5.75% | FeNaEDTA |
| S | 7.48% | 5.75% | $MgSO_4$ |
| Mg | 5.67% | 5.75% | $MgSO_4$ |
| Na | 1.08% | 1.16% | FeNaEDTA, $Na_2MoO_4 \cdot 2H_2O$ |
| Cl | 0.85% | 0.59% | $MnCl_2 \cdot 4H_2O$, $ZnCl_2$, $CuCl_2 \cdot 2H_2O$ |
| B | 0.584% | 1.330% | $H_3BO_3$ |
| Fe | 1.303% | 0.575% | FeNaEDTA |
| Mn | 0.586% | 0.263% | $MnCl_2 \cdot 4H_2O$ |
| Zn | 0.062% | 0.023% | $ZnCl_2$ |
| Cu | 0.022% | 0.008% | $CuCl_2 \cdot 2H_2O$ |
| Mo | 0.012% | 0.003% | $Na_2MoO_4 \cdot 2H_2O$ |

four sampling times (0, 5, 60 and 180 minutes). The short-term timeframe of this experiment was chosen to ensure that intact uptake of the originally labelled compound was the dominant factor in observing its $^{15}$N signal in plant tissue.

## Plant labelling and processing

All plants were dosed with a N solution containing a mixture of $NO_3^-$, $NH_4^+$, urea and alanine. One of the four N compounds was isotopically-labelled in each treatment, with a fifth control treatment containing no labelling (Table 2). The four N compounds were supplied in the same concentration to every seedling, proportionally to the three bioavailable N pools found in intensive agricultural environments: ~33% inorganic N ($NO_3^-$ and $NH_4^+$), ~33% DON (alanine), and ~33% urea N (urea).

Once plants reached a two- to four-leaf stage, the rhizotube pots were injected with 2 mL of solution at three points around the root base, with special care taken to avoid injecting into any roots (S1 Fig). Plants were then placed in direct sunlight for their allocated uptake period (5, 60 and 180 minutes). After the treatment times had elapsed, plants were carefully extracted from their pots, thoroughly washed of substrate, bagged, and placed on dry ice ($CO_{2(s)}$) to halt metabolism. Plants were transferred to ovens to air dry (at 40°C) completely at the end of each day. Dried samples were weighed and ground to a fine powder using a ball-mill. Ground samples were analysed for $\delta^{13}$C and $\delta^{15}$N using isotope ratio mass spectrometry (IRMS) at the Research School of Biology at the ANU, Canberra, ACT. Total C and N were measured on an Elementar VarioMAX CNS elemental analyser.

## Calculation of $^{13}$C and $^{15}$N uptake

The process of transposing relative $\delta^{15}$N and $\delta^{13}$C values to absolute percentage uptake values followed Equations 1–4. All equations listed below were used for both N and C calculations, with the $^{15}$N and $^{14}$N equation pronumerals substituted with $^{13}$C and $^{12}$C respectively in the C calculations.

The calculation of mass fraction (X) from relative isotopic enrichment ($\delta$) where $R(^{13}C)_{standard} = 0.011178$ and $R(^{15}N)_{standard} = 0.00367647$ followed equation 1.

$$X(^{15}N)_{sample} = \frac{R(^{15}N)_{standard} \cdot \left(\frac{\delta^{15}N}{1000} + 1\right)}{1 + \left(R(^{15}N)_{standard} \cdot \left(\frac{\delta^{15}N}{1000} + 1\right)\right)}$$

(1)

Table 2. Nitrogen treatments for each of the three G. hirsutum varieties included in this study. The total N concentration of all solutions was 2.95 mmol N L-1, with approximately 1 mmol N L-1 apportioned to each of the three plant-available N pools outlined in this study: mineral N, synthetic N and DON.

| | N Treatment | Nitrate | Ammonium | Urea | Alanine |
|---|---|---|---|---|---|
| Plant-Available N Pool: | | Mineral N | Mineral N | Synthetic N | LMW DON |
| Nitrogen Isotope Purity: | | 99.8% $^{15}$N | 99.5% $^{15}$N | 99% $^{15}$N | 99% $^{15}$N |
| Carbon Isotope Purity: | | – | – | 98% $^{13}$C | 99% $^{13}$C |
| Concentration (mmol N L$^{-1}$): | | 0.61 | 0.35 | 1.00 | 1.00 |
| Concentration (mmol C L$^{-1}$): | | – | – | 0.50 | 3.00 |
| | 0 | Unlabelled | Unlabelled | Unlabelled | Unlabelled |
| | 1 | $^{15}$N-Labelled | Unlabelled | Unlabelled | Unlabelled |
| | 2 | Unlabelled | $^{15}$N-Labelled | Unlabelled | Unlabelled |
| | 3 | Unlabelled | Unlabelled | $^{15}$N-$^{13}$C-Labelled | Unlabelled |
| | 4 | Unlabelled | Unlabelled | Unlabelled | $^{15}$N-$^{13}$C-Labelled |

The calculation of percentage N derived from transfer (%NDFT) from mass fractions (X) of sample, isotope label, and cultivar variety control where $X(^{15}N)_{label} = \%{}^{15}N\ Purity_{label}$ and $X(^{15}N)_{variety\ control} = (\sum X(^{15}N)_{control}) \div n$, where n = number of replicates (always 10 in this experiment) followed equation 2.

$$\%NDFT = \frac{\%\ ^{15}N\ Atom\ Excess_{sample}}{\%\ ^{15}N\ Atom\ Excess_{donor}} \times 100\% = \frac{X(^{15}N)_{sample} - X(^{15}N)_{variety\ control}}{X(^{15}N)_{label} - X(^{15}N)_{variety\ control}} \times 100\%$$

(2)

The calculation of the mass of fertiliser $^{15}N$ taken up by the plant using %NDFT, plant dry mass (grams), and plant percentage N followed equation 3.

$$m(Fert.\ ^{15}N)_{taken\ up} = \%NDFT \times m(Plant)_{dry} \times Plant\ \%N$$

(3)

The calculation of percentage fertiliser $^{15}N$ uptake by the plant using molar mass ($M_r$) of $^{15}N$ (15 mg mmol$^{-1}$), percentage purity (%Purity) of isotope labels (98.0–99.8%), N concentration (mmol L$^{-1}$), and volume (L) followed equation 4.

$$\%\ Fert.\ ^{15}N\ Uptake = \frac{m(Fert.\ ^{15}N)_{taken\ up}}{m(Fert.\ ^{15}N)_{added}} = \frac{m(Fert.\ ^{15}N)_{taken\ up}}{[Label-N] \times \%Purity \times V_{fert.sol.} \times M_r(^{15}N)}$$

(4)

The calculation of the fraction of the plant uptake $^{15}N$ to the initial solution $^{15}N$ was determine using the $^{15}N$ content in the plant and the initial solution followed equation 5.

$$Fraction\ of\ initial = \frac{Fertiliser\ ^{15}N\ \ uptake}{^{15}N\ \ Intital\ solution}$$

(5)

### Statistical analysis

All statistical analyses were performed in R v3.6 [26] using the lme4 linear mixed effects modelling package [27] was used to determine the preference of N in the different varieties.

## Results

### Growing phase

The washed and pasteurised sand used as the growing medium total N (0.003% ± 0.0001) and total C (0.029% ± 0.001) concentrations were negligible prior to planting. The three *G. hirsutum* varieties did not exhibit uniform germination characteristics (S2 Fig). However, by the date of labelling, the Sicot, Sicala and Tx varieties achieved germination rates of 88%, 97% and 86% respectively.

### Nitrogen compound uptake

All three *G. hirsutum* varieties rapidly utilised the added $NO_3^-$, $NH_4^+$, urea ($CH_4N_2O$) and alanine ($C_3H_7NO_2$; Fig 1). *G. hirsutum* varieties showed a small preference for inorganic N ($NO_3^-$ and $NH_4^+$) over organic N (alanine and urea), with respective mean uptakes of added N 20.9 ± 0.04% and 18.3 ± 0.04% for the two N pools (Table 2).

### Timing of nitrogen uptake

Uptake of all N species presented as linear over the 180-minute experiment window, suggesting zero order kinetics over this timeframe. Supplied N was taken up at rates of 4.6–8.1% of added fertiliser N per hour, equating to 0.28–2.11 mg N hour$^{-1}$ (Tables 3 and 4).

 

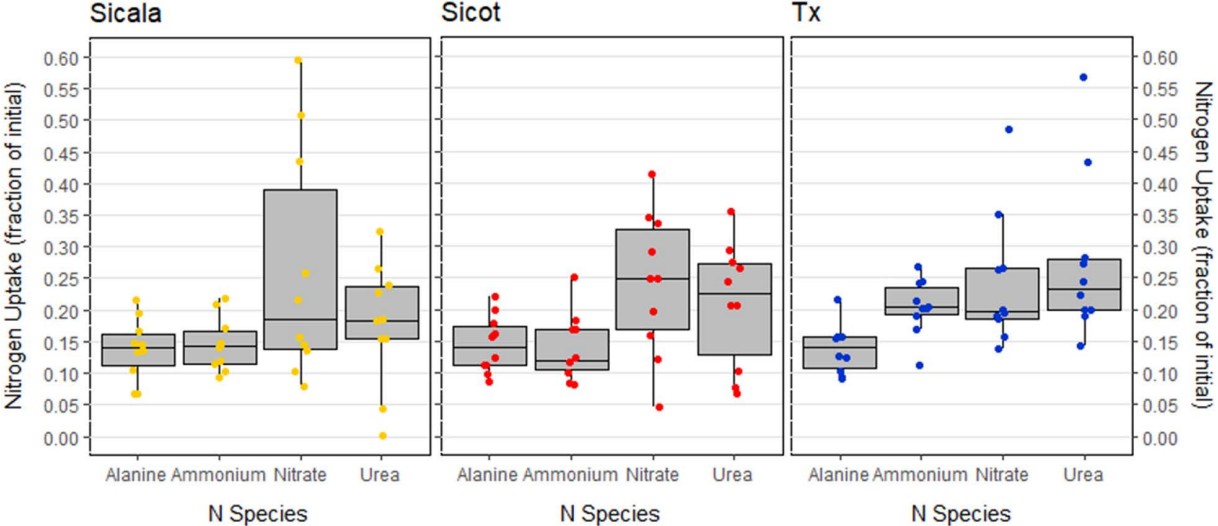

**Fig 1. N uptake after 180 minutes (nominal) for each *G. hirsutum* variety: Sicot 746B3F, Sicala V2 and Tx III.** The four N compounds $NO_3^-$, $NH_4^+$, urea ($CH_4N_2O$) and alanine ($C_3H_7NO_2$) were added concurrently.

**Table 3. *G. hirsutum* estimated marginal mean uptake of N species after 180 minutes of exposure. Values are mean ± standard error, n = 10.**

| Nitrogen Species | % uptake of the added N |
| --- | --- |
| Alanine | 14.3 ± 1.9% |
| Ammonium | 16.6 ± 1.9% |
| Nitrate | 25.1 ± 1.9% |
| Urea | 22.2 ± 1.9% |
| Inorganic N ($NO_3^-$ + $NH_4^+$) | 20.9 ± 3.8% |
| Organic N (alanine + urea) | 18.3 ± 0.38% |

**Table 4. Linear N uptake rate of each fertiliser species for average of all three *G. hirsutum* varieties (n = 90).**

| | Uptake Rate (% uptake per hour) | Uptake Rate (Hours for 100% Uptake) | Fit ($r^2$) |
| --- | --- | --- | --- |
| Alanine | 4.6% | 21.7 | 0.997 |
| Ammonium | 5.4% | 18.6 | 0.997 |
| Nitrate | 8.1% | 12.3 | 0.994 |
| Urea | 7.1% | 14.1 | 0.973 |

## Cotton variety

Neither total (Fig 1B) nor N compound uptake (Fig 2) were statistically different between the three *G. hirsutum* varieties (p = 0.20 and p = 0.47 respectively). However, differences in total N uptake across the different N compounds (Fig 2B) were significant (p < 0.001). After 180 minutes, the uptake of $NH_4^+$–N and alanine—N were not statistically different (p = 0.77). Neither were $NH_4^+$–N and urea—N (p = 0.11).

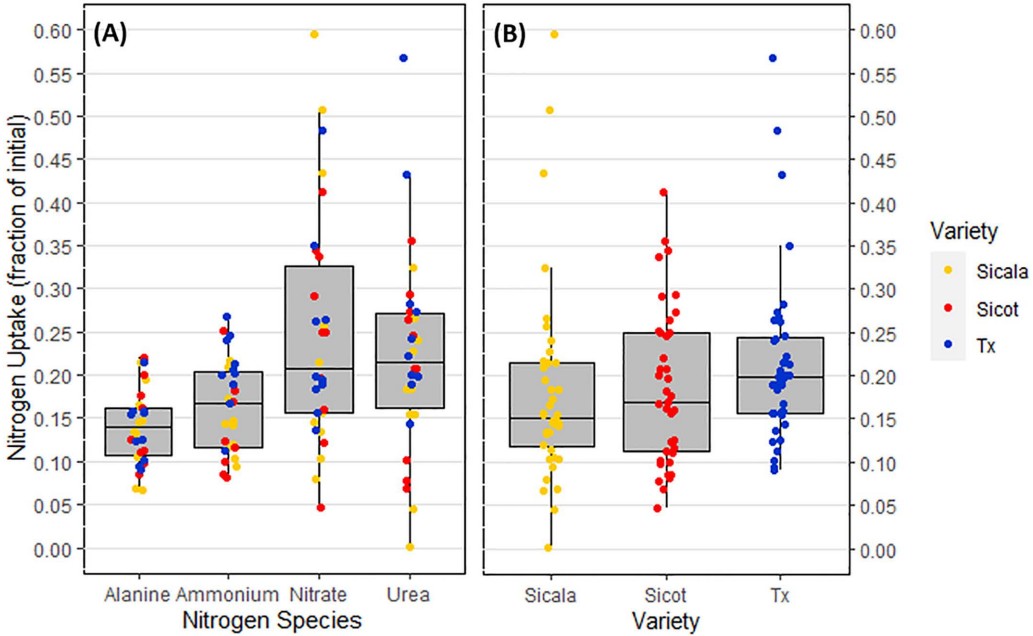

**Fig 2. Nitrogen uptake after 180 minutes (nominal). (A)** Average N compound uptake for all varieties. **(B)** Total N uptake for each *G. hirsutum* variety.

The uptake of alanine—N and urea—N represented the only N compound uptake difference within each varieties and were statistically different in the landrace accession Tx III (p < 0.05) but not the two commercial cultivars (p = 0.58). Uptake of $NO_3^-$–N and alanine—N, and $NO_3^-$–N and $NH_4^+$–N were both statistically different across all varieties (p < 0.01), while $NO_3^-$–N and urea—N were not (p = 0.63).

### Nitrogen and carbon uptake

Alanine—C was taken up linearly over the 180-minute window by all *G. hirsutum* varieties, similarly to alanine—N (Fig 3). There were no statistically significant differences in alanine—C or alanine—N uptake between the three *G. hirsutum* varieties (p = 0.73). C and N were taken up in a consistent ratio of approximately 0.32:1 (Fig 4).

The uptake of urea—C is not linear over the 180-minute experiment window (Fig 5). Over the full 180 minutes, no correlation was found between urea—C uptake and variety or urea—N uptake (p = 0.96; Fig 6). Internal plant urea—C concentration rose over the first 60 minutes, and then fell back to almost zero by the end of the 180 minutes.

## Discussion

### Direct plant access of soil organic nitrogen

Our results demonstrate that LMW DON can be a direct source of nutrition for commercial cotton (*G. hirsutum*). In some situations, this may comprise a meaningful portion of the cotton plant's uptake of N in field situations. Recent studies have shown that cotton obtains the majority of its N from the soil N pool [28] and that organic N is a major component of soil N in most agricultural soils [15,29]. However, as this experiment was not designed to emulate real-world conditions, further soil-based glasshouse and *in situ* experiments will be required determine if this is the case in more complex non-sterile systems.

*G. hirsutum* does not appear to exhibit any uptake preferences between the four N species added. All four N species were taken up concurrently, suggesting that cotton is not particularly selective about the form in which it receives N. All

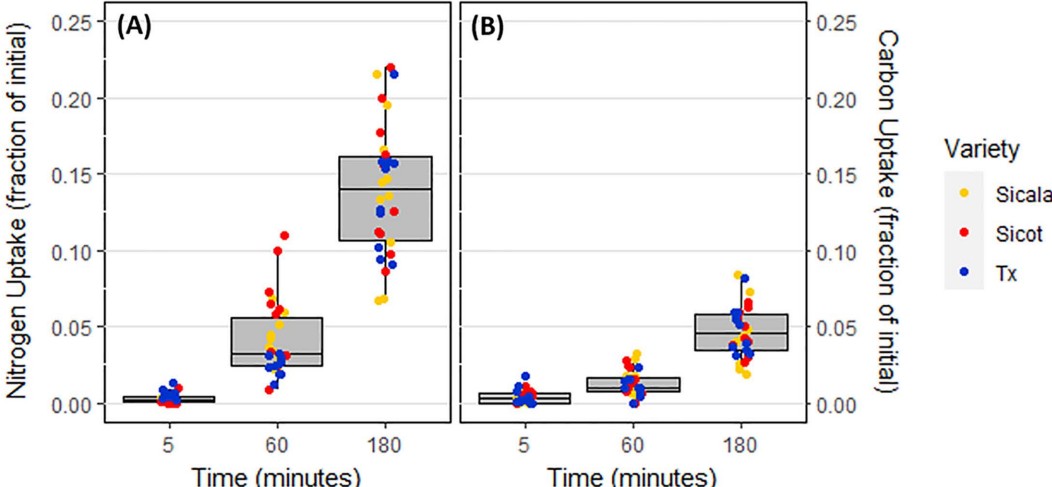

**Fig 3. Average alanine uptake of all _G. hirsutum_ varieties within the three sampling intervals. (A)** Alanine—N uptake. **(B)** Alanine—C uptake.

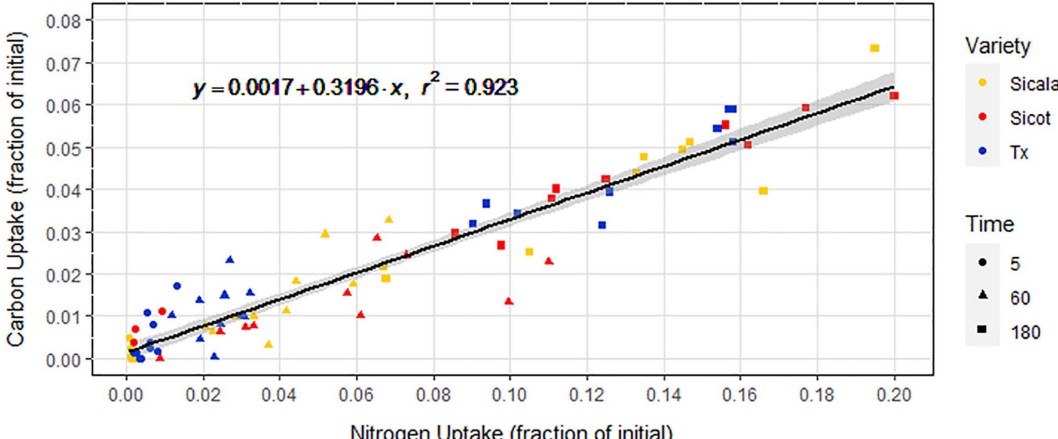

**Fig 4. Alanine—C uptake against alanine—N uptake plotted as fraction of initial concentration.** Equation describes the linear regression that is plotted in black with 95% confidence interval in grey. Data point shapes denote isotope exposure times in minutes. The uptake of alanine—C and alanine—N are highly correlated ($r^2 = 0.92$, $p < 0.001$).

three _G. hirsutum_ varieties exhibited a small preference for inorganic N ($NO_3^-$ and $NH_4^+$) over organic N (alanine and urea), with mean total uptake of the two N pools 20.9±0.04% and 18.3±0.04% of added N respectively. These research findings are consistent with those observed in Australian sugarcane (_Saccharum officinarum_ L.), which is also produced in high-input systems that are comparable to Australian irrigated cotton. Sugarcane has been shown to readily take up amino acids from the soil solution, which can constitute a significant source of N for commercial crops [30,31].

The _G. hirsutum_ plants used in this experiment were grown from seed in washed and pasteurised sand. While this may not have completely eliminated mycorrhizal fungi or other soil microorganisms from the growing environment, it will most likely have significantly diminished their presence and role in plant N uptake. This experiment did not test for mycorrhizal activity, but it is a credible assumption that the _G. hirsutum_ seedlings took up the majority of the alanine—N and urea—N without the aid of symbiotic mycorrhizal fungi [20,32,33].

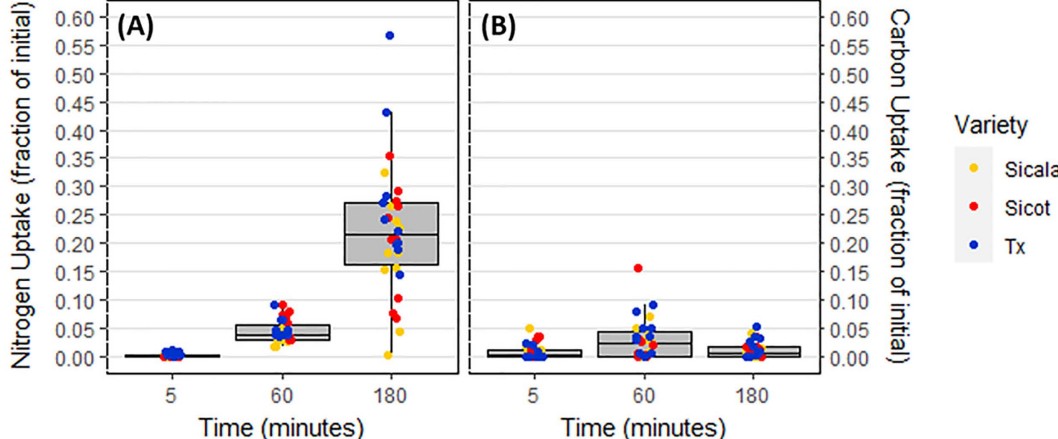

**Fig 5. Average urea uptake of all *G. hirsutum* varieties over the three sampling intervals. (A)** Urea—N uptake. **(B)** Urea—C uptake, which does not increase consistently over time.

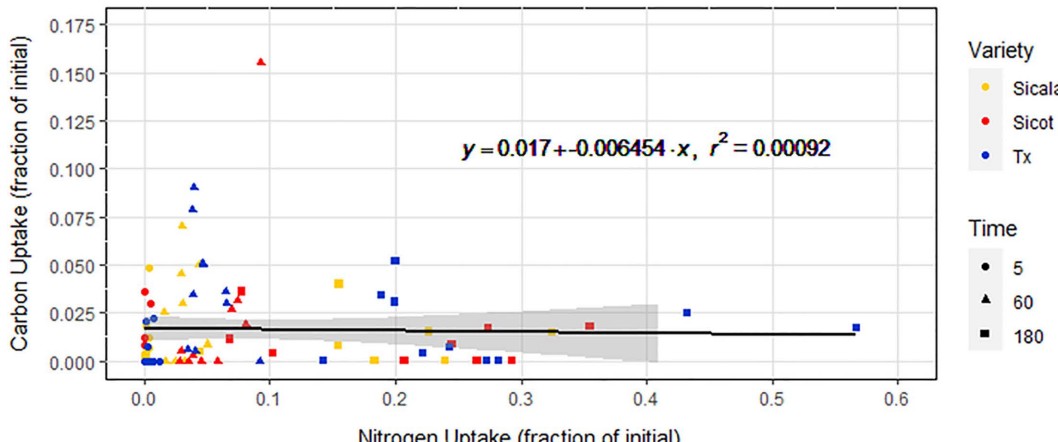

$$y = 0.017 + -0.006454 \cdot x, \ r^2 = 0.00092$$

**Fig 6. Urea—C uptake against urea—N uptake plotted as fraction of initial concentration.** Equation describes the linear regression that is plotted in black with 95% confidence interval in grey. Data point shapes denote isotope exposure times in minutes. Urea—N and urea—C uptake do not correlate over the whole 180-minute experimental window ($r^2 < 0.001$, $p = 0.96$).

In recent years, studies have shown that a wide variety of plants can take up organic N, including: trees [34–41], grasses and sedges [30,31,42–44], mosses and lichens [43,45,46], fruits [47,48], and broadacre crops including wheat, maize, chicory, and lupin [49–51]. This list of plants known to take up organic N can now be expanded to include commercial cotton (*G. hirsutum*), a high-value, high-input pillar crop. Given the large proportion of N taken up by cotton crops that does not come directly from fertiliser [11], and the larger proportion of N present as DON than previously recognised [52], this implies that direct uptake of organic N may be an important pathway of N nutrition in cotton.

### Organic carbon uptake mechanism

Similarly to alanine—N, alanine—C was taken up linearly over the 180-minute window by all *G. hirsutum* varieties. The consistent internal seedling $^{13}C:^{15}N$ ratio of 0.32:1 suggests that approximately 32% of the alanine—N taken up was in the

form of whole alanine molecules. Many other studies have already demonstrated the intact uptake of the LMW molecules alanine [25], arginine [39], glycine [53], acetate [49] and trialanine [54], as well as plant consumption of whole proteins and microbes [33,55,56], but never before in *G. hirsutum*. While it is possible that the alanine—C taken up was not in the form of intact alanine, functionally this is indistinct from intact alanine uptake (Fig 7). The majority of alanine—N taken up was not associated with alanine—C, suggesting that extracellular deamination is the dominant alanine—N uptake pathway. It is also possible that all absorbed alanine—N was as whole alanine molecules, and that approximately 68% of the alanine—C was subsequently expelled post-intracellular deamination. Regardless, these results suggest that at least some alanine was taken up whole.

In the early stages of growth, much N taken up from the soil is typically transported to the leaves to create proteins like Ribulose-1,5-bisphosphate carboxylase/oxygenase (Rubisco; [57,58]). Taking up intact amino acids and transporting them whole may provide a biochemical energy advantage over taking up inorganic N and assembling amino acids from it. This should be explored further, with previous studies already confirming that high organic N uptake can alter internal N distribution within plants [31].

In contrast to the behaviour of alanine—C, urea—C uptake did not follow the same linear trend observed in the other N and C species. Internal plant urea—C concentration increasing over the first 60 minutes then subsequently fell to almost zero by 180 minutes. This was true for all three *G. hirsutum* varieties, with no significant differences between them (p = 0.34). One possible mechanistic explanation for this behaviour is the intact uptake of some urea, followed by rapid intracellular ammonification and subsequent expulsion of the urea—C as $CO_2$. While the dominant urea—N uptake pathway appears to involve extracellular ammonification, the apparent initial uptake of some intact urea by *G. hirsutum* requires further investigation. If it is demonstrated more widely, this could provide options to target higher fertiliser—N capture by the crop through improved breeding which may reduce N loss to the environment.

## Conclusion

This study has demonstrated for the first time that *G. hirsutum* can rapidly and concurrently access the DON, inorganic N and synthetic N pools in soil. As global fibre production—mostly comprising of cotton—is responsible for 4.3% of the global fertiliser N budget, opportunities to reduce fertiliser requirements whilst maintaining productivity and not mining soil organic matter are of significance. While all N species were taken up, total N uptake from each N compound was different. *G. hirsutum* exhibited a small preference for inorganic N ($NO_3^-$ and $NH_4^+$) over organic N (alanine and urea). The consistent uptake ratio of alanine—C to —N of 0.32:1 indicates that some alanine is likely taken up intact by the plant, whereas no correlation was found between urea—C uptake and urea—N uptake indicating that urea undergoes rapid extracellular ammonification prior to uptake of its liberated inorganic N. Interestingly, neither total nor speciated N uptake were statistically different between the three *G. hirsutum* varieties. As our study included three different *G. hirsutum* varieties ranging from the landrace Tx III representing the native origins of the commercial *G. hirsutum* through to a modern-day elite GM cultivar, it appears that the physiological N uptake preference observed in this study is conserved across varieties. The implications of this observation are that breeding does not appear to have altered the physiological N preferences of *G. hirsutum*. Although it is unlikely that N uptake preference has been a target trait for breeders, this does suggest limited plasticity in physiological N preference. Further research is needed to assess if the physiological preferences shown here also occur in *G. hirsutum* grown in soil and thus in competition with the microbial community and other biogeochemical processes.

## Supporting information

**S1 Fig. Processing of *G. hirsutum* seedlings in $^{15}N^{13}C$ uptake glasshouse experiment. (A)** Injection of rhizotube with N solution. Careful attention was paid to not inject solution directly into the plant. **(B)** Extracted and cleaned *G. hirsutum* seedling. Once cleaned, seedlings were bagged and placed on dry ice ($CO_{2(s)}$) to halt metabolism.
(DOCX)

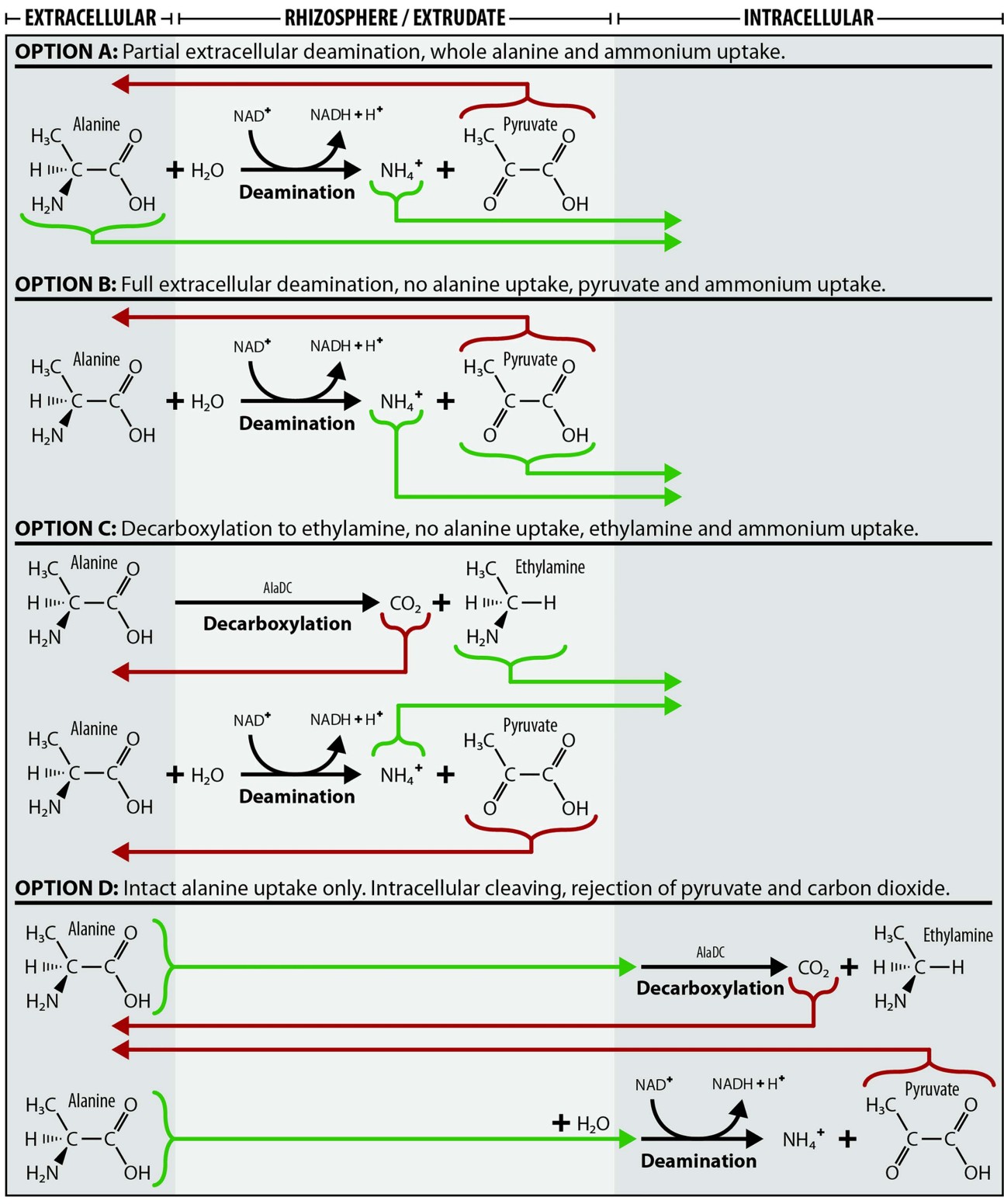

**Fig 7. Four potential alanine—N and –C uptake scenarios.** Experimental results suggest the majority of alanine—N is taken up as an amine functional group, potentially $NH_4^+$, which would likely mean extracellular deamination. Alanine—C uptake suggests that up to 32% of alanine—N was taken up as whole alanine molecules. Actual uptake behaviour may be a combination, or none, of these options. AlaDC is an abbreviation of the enzyme alanine decarboxylase.

**S2 Fig. Germination percentage for each of the three G. hirsutum varieties: Sicot 746B3F, a GM current commercial cultivar; Sicala V2, an obsolete non-GM commercial cultivar; and Tx III, a Guatemalan landrace accession.** n = 720 seeds.
(DOCX)

## Acknowledgments

The authors gratefully acknowledge technical assistance from the ANU Research School of Biology IRMS facility for sample analysis.

## Author contributions

**Conceptualization:** James O Latimer, Mark Farrell.

**Data curation:** James O Latimer.

**Formal analysis:** James O Latimer, Mark Farrell.

**Funding acquisition:** Mark Farrell, Ben CT Macdonald.

**Investigation:** James O Latimer, Mark Farrell, Ben CT Macdonald.

**Methodology:** James O Latimer, Mark Farrell, Ben CT Macdonald.

**Project administration:** Mark Farrell.

**Resources:** Mark Farrell, Ben CT Macdonald.

**Supervision:** Mark Farrell, Ben CT Macdonald.

**Visualization:** James O Latimer, Mark Farrell.

**Writing – original draft:** James O Latimer, Mark Farrell.

**Writing – review & editing:** James O Latimer, Mark Farrell, Ben CT Macdonald.

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
