## [Decision Letter · Decision Letter 0]

14 Mar 2025

Dear Dr. Farrell,

Thank you for submitting your manuscript to PLOS ONE. After careful consideration, we feel that it has merit but does not fully meet PLOS ONE’s publication criteria as it currently stands. Therefore, we invite you to submit a revised version of the manuscript that addresses the points raised during the review process.

We look forward to receiving your revised manuscript.

Kind regards,

Narendra Khatri, Ph.D.

Academic Editor

PLOS ONE

Journal Requirements:

5. We note you have included a table to which you do not refer in the text of your manuscript. Please ensure that you refer to Table 4 in your text; if accepted, production will need this reference to link the reader to the Table.

Additional Editor Comments:

The authors are advised to revise the manuscript according to the reviewers' comments and resubmit it for further review.

Reviewers' comments:

Reviewer's Responses to Questions

**Comments to the Author**

1. Is the manuscript technically sound, and do the data support the conclusions?

Reviewer #1: Yes

Reviewer #2: Yes

2. Has the statistical analysis been performed appropriately and rigorously?

Reviewer #1: No

Reviewer #2: Yes

3. Have the authors made all data underlying the findings in their manuscript fully available?

Reviewer #1: Yes

Reviewer #2: Yes

4. Is the manuscript presented in an intelligible fashion and written in standard English?

Reviewer #1: No

Reviewer #2: Yes

Reviewer #1: The manuscript entitled “Nitrogen uptake preference of Gossypium hirsutum L. from a mixture of urea, alanine, nitrate, and ammonium" has been reviewed. The authors provided the information on the ability of G. hirstutum to take up different organic and inorganic N forms in a sand matrix. It is well known that plants always take nitrogen in inorganic form. The manuscript, at least as written, lacks novelty. Despite the fact that, I provided some points to the authors which are address below. I have also highlighted and commented some points in Manuscript.

• The English in the paper is very rough and requires the assistance of a more practiced English expert to make the paper more easily read and understandable.

• The abstract still requires substantial editing.

• The Purpose of the study in the abstract section is not clear. It should be concise within two to three lines.

• Conclusion part in abstract section needs to be rewrite, it should be clear for better understanding of the readers

• The quality of writing should be improved.

• Introduction section is deficient. The authors did not present a novel justification for carrying out this study.

• There is a dire need to address grammatical errors, as the sentence structure is unclear and confusing. This will make it difficult for readers to grasp the intended meaning.

• The results and discussion section should be improved, well-structured and interconnected and the discussion should be link to all your findings.

• The authors should finish the discussion with what are the main points (take home massage).

• The paragraphs in result discussion section seems to be correspond to the results only and not to a real discussion of your results. The paragraphs are not appropriately discussed.

• The deeper scientific interpretations of your findings are strongly suggested.

• The conclusion must be self-explanatory and highlight novelty and implication of your study.

• Make sure that all figures and tables are cited within the text and that they are cited in consecutive order.

• References should be in Format

• Please use Some recent references

Reviewer #2: The study aims to decipher N uptake preference from N sources viz. urea, alanine, nitrate, and ammonium through isotopic studies and probable mechanism thereof. The methodology is robust and all the results pertaining to the objective are elaborated. The paper may be accepted after minor revisions as mentioned.

**Do you want your identity to be public for this peer review?** For information about this choice, including consent withdrawal, please see our Privacy Policy

Reviewer #1: No

Reviewer #2: **Yes: ** Immanuel Chongboi Haokip

---

## [Author Response · Author response to Decision Letter 1]

28 Apr 2025

Response to reviewers – PONE-D-24-59331

We thank the reviewers and editorial time in handling and reviewing our submission to PLoS1. We are pleased to address these comments, and our responses are given below in blue. As requested, we have also submitted a version of the MS with changes tracked to better facilitate the review process. All line numbers in our responses refer to the line numbers in the revised, clean version.

Editorial comments:

Journal Requirements:

Thank you for highlighting these issues. We apologies for not having thoroughly attended to these issues prior to the first submission, and hope that they are now deemed addressed satisfactorily. We will be happy to make any further modifications as required to meet the Journal’s specifications.

This is now done

All authors have agreed to this statement. Publication of data relating to publications on the CSIRO Data Access Portal (data.csiro.au) is routine for publications arising from CSIRO authors.

Apologies. This has now been completed by all authors.

5. We note you have included a table to which you do not refer in the text of your manuscript. Please ensure that you refer to Table 4 in your text; if accepted, production will need this reference to link the reader to the Table.

Apologies, this is now referenced at L169

Additional Editor Comments:

The authors are advised to revise the manuscript according to the reviewers' comments and resubmit it for further review.

Reviewer 1:

The manuscript entitled “Nitrogen uptake preference of Gossypium hirsutum L. from a mixture of urea, alanine, nitrate, and ammonium" has been reviewed. The authors provided the information on the ability of G. hirstutum to take up different organic and inorganic N forms in a sand matrix. It is well known that plants always take nitrogen in inorganic form. The manuscript, at least as written, lacks novelty. Despite the fact that, I provided some points to the authors which are address below. I have also highlighted and commented some points in Manuscript.

We thank the reviewer for their comments. We contest their statement that “it is well known that plants always take nitrogen in inorganic form” as this is an inaccurate viewpoint that has been superseded over the past three decades of research into organic nitrogen nutrition of many species of higher plants (e.g., grasses, forbs, shrubs, trees, cereals), demonstrating their capacity to access organic nitrogen in the form of amino acids, short peptides, quaternary ammonium compounds, and even potentially whole proteins and microorganisms. Our data clearly demonstrate for the first time that the economically important G. hirsutum can now be added to that list.

• The English in the paper is very rough and requires the assistance of a more practiced English expert to make the paper more easily read and understandable.

All three authors are native English speakers. In redrafting this revision, we have made adjustments where extra clarity was required, and have also addressed the specific points raised by this reviewer.

• The abstract still requires substantial editing.

We have revised the abstract in line with the specific comments below from this reviewer

• The Purpose of the study in the abstract section is not clear. It should be concise within two to three lines.

We have clearly stated the purpose in response to this request

• Conclusion part in abstract section needs to be rewrite, it should be clear for better understanding of the readers

We have now re-arranged and expanded the concluding remarks of the abstract

• The quality of writing should be improved.

Please see earlier response

• Introduction section is deficient. The authors did not present a novel justification for carrying out this study.

We emphasise that the ability of the commercially important G. hirsutum to access organic N has not been previously examined at L75.

• There is a dire need to address grammatical errors, as the sentence structure is unclear and confusing. This will make it difficult for readers to grasp the intended meaning.

We have addressed the specific points on clarity raised by this reviewer below.

• The results and discussion section should be improved, well-structured and interconnected and the discussion should be link to all your findings.

Separating the results and discussion sections is preferred as this allows clear distinctions to be drawn between the findings of the study (results section) and their interpretation in light of published studies (discussion section). Specific comments raised by this reviewer have been addressed below.

• The authors should finish the discussion with what are the main points (take home massage).

The discussion is structured into sub-sections, as is common in scientific writing. We have taken care to ensure that each subsection now clearly emphasises its take-home message.

• The paragraphs in result discussion section seems to be correspond to the results only and not to a real discussion of your results. The paragraphs are not appropriately discussed.

As per our earlier response, the purpose of separating the results and discussion sections enables a delineation between the findings of the study (results section) and its implications in the context of the wider literature (discussion section).

• The deeper scientific interpretations of your findings are strongly suggested.

This study targeted examination of the physiological capacity of G. hirsutum to take up organic N forms, rather than questioning whether it does so in competition with soil microorganisms in field conditions. We highlight this limitation at L197. Thus, whilst we have modified text throughout the discussion, we do not feel it appropriate to over-interpret these data.

• The conclusion must be self-explanatory and highlight novelty and implication of your study.

We agree that the conclusion was not satisfactory and have completely revised this section.

• Make sure that all figures and tables are cited within the text and that they are cited in consecutive order.

This has now been checked

• References should be in Format

We have used the PLoS format in EndNote, and this appears to match that of the PLoS One guidelines

• Please use Some recent references

Two recent references (Gao et al 2025 #23, Bailey et al 2022 #51) have been added

L1 Improve the title

We have now simplified the title

L19 Rewrite the sentence. Not clear

Now done

L37 Sentenceis not clear. Not understanding the meaning of sentence.

Now revised for clarity

L59 Very big sentence, not clear to understand, some words are missing, role of microbes. please try to read and rewrite again.

Now revised for clarity

L126 Write in seperate line for more clear picture and understandin

Now revised for clarity

L135 two times solution please write the sentence clearly

Now revised for clarity

L136 Superscript

Thank you - fixed

L241 Check and rewrite

Sentence has been removed as part of the broader re-write and improvements of the conclusions section

Reviewer 2:

The study aims to decipher N uptake preference from N sources viz. urea, alanine, nitrate, and ammonium through isotopic studies and probable mechanism thereof.

The methodology is robust and all the results pertaining to the objective are elaborated. The paper may be accepted after minor revisions as mentioned below:

We thank the reviewer for their supportive and constructive comments

Check the subheadings. Some subheadings are in running sentences while in some, each words are capitalized .

This has now been resolved

Abbreviations used in the paper must be described at the first mention and used uniformly thereafter.

This has now been checked and resolved. The only exception to this is in discussion of the results where we do use NH4+--N and NO3--N to be accurate in the comparison with the N from the organic N cpds (e.g. alanine-N) as of course we only track the 15N and not the entire molecule. Thus we are being specific in talking about the N (or C) from each source molecule.

Line number 45: check spelling of minerialisation

Fixed, thank you

Line number 98-100: Mention the statistical design

Now done, thank you

Line number 111: (s) should be normal font size?

Fixed, thank you

Line number 123, 127….: Equation may be presented a (1)

Now done, thank you

Line number 139: R version may be mentioned

Now done, thank you

In some, species are mentioned as NH4+–N and alanine—N while in some cases NH4+, alanine, etc are used. Uniform names may be followed.

We explicitly highlight the –N in these cases when discussing the results as the isotopic analysis only allows us to be absolutely certain about the N (or C) and not the rest of the molecule in question. So, to treat N from nitrate or N from alanine equally, we must emphasise that it is only the N that is being discussed and not the rest of the molecule (that can only be presumed to have followed).

References may be double-checked to follow the journal referencing guidelines and follow uniform referencing style throughout. Double-check to make sure that all references have a corresponding citation within the text and vice versa.

This has now been done. We used the Endnote style for PLOS and made minor modifications to this to match current formatting instructions on the PLOS1 website.

---

## [Decision Letter · Decision Letter 1]

23 Jun 2025

Dear Dr. Farrell,

Thank you for submitting your manuscript to PLOS ONE. After careful consideration, we feel that it has merit but does not fully meet PLOS ONE’s publication criteria as it currently stands. Therefore, we invite you to submit a revised version of the manuscript that addresses the points raised during the review process.

We look forward to receiving your revised manuscript.

Kind regards,

Roshan Babu Ojha

Academic Editor

PLOS ONE

Journal Requirements:

Additional Editor Comments:

I thank authors for carrying out this exciting work, however, there still some correction is necessary.

There is no rationale why only alanine and urea as an organic N form is selected as treatment because there are many other organic N forms are available? It is somehow reflected in hypothesis, but a clear statement would help.

You have used labelled N and C to exactly quantify the N uptake by cotton, I suggest making a graphical abstract showing how much proportion of N uptake through several N forms.

L123, How the 13C information used in N uptake (equation 1-4)?

L188, present r coeff value. Is this a no correlation or non-significant correlation?

L264, L266, Fix syntax error (eg space and spelling)

This is 180 minutes research suggesting a N uptake preference of cotton. Please justify why only 180 minutes, people working on isotope might understand but it is good to provide some rationale to clarify wider audience.

Overall, this an intensive work and appreciate authors efforts.

Reviewers' comments:

Reviewer's Responses to Questions

**Comments to the Author**

Reviewer #2: All comments have been addressed

2. Is the manuscript technically sound, and do the data support the conclusions?

Reviewer #2: Yes

3. Has the statistical analysis been performed appropriately and rigorously?

Reviewer #2: Yes

4. Have the authors made all data underlying the findings in their manuscript fully available?

Reviewer #2: Yes

5. Is the manuscript presented in an intelligible fashion and written in standard English?

Reviewer #2: Yes

Reviewer #2: Line no. 2: Is BCT the abbreviation for Bennet C.T.? If yes, is it suitable to remove BCT?

Line no. 21: Two full stops, remove one of them

Line no. 28: What do you mean by taking up intact? Do you mean to say that they are taken up in compound form without any dissociation?

Line no. 72: DON?? Abbreviate it first

Line no. 74: inconcert or in concert?

Line no. 41: compare the dash used in line 41, 46,142, 169 similar dashes may be used

Line no. 117: give space between 40 and °C

Line no. 193: check spelling of demomnstrate

Line no. 196: soil N pool or soil organic n pool?

Line no. 264: add space between G. hirsutum and through

Line no. 271: correct the spelling of biogeochmaical

In references, the number is repeated. Remove one of them. The format needs to be revised, for example, some of the journal name are in abbreviation while some are in full, some doi is presented as doi: while some are in https://doi.org/ , see the journal format again.

**Do you want your identity to be public for this peer review?** For information about this choice, including consent withdrawal, please see our Privacy Policy

Reviewer #2: **Yes: ** Immanuel Chongboi Haokip

---

## [Author Response · Author response to Decision Letter 2]

30 Sep 2025

Reviewer #2:

• Line no. 2: Is BCT the abbreviation for Bennet C.T.? If yes, is it suitable to remove BCT? [BCT removed]

• Line no. 21: Two full stops, remove one of them [removed]

• Line no. 28: What do you mean by taking up intact? Do you mean to say that they are taken up in compound form without any dissociation? [have added “…was absorbed intact without extracellular deamination.”]

• Line no. 72: DON?? Abbreviate it first [It was defined on line 58]

• Line no. 74: inconcert or in concert? [space inserted]

• Line no. 41: compare the dash used in line 41, 46,142, 169 similar dashes may be used [These dashes all converted to standard “-“ dash. All em dashes, en dashes and hyphens have been reviewed in the document

Em dash (—) e.g. alanine—N

En dash (–) e.g. NH4+–N

Hyphen (-) e.g. 40-50

]

• Line no. 117: give space between 40 and °C [space inserted]

• Line no. 193: check spelling of demomnstrate [corrected]

• Line no. 196: soil N pool or soil organic n pool? [reworded for clarity]

• Line no. 264: add space between G. hirsutum and through [corrected]

• Line no. 271: correct the spelling of biogeochmaical [corrected]

Additional Editor Comments:

• There is no rationale why only alanine and urea as an organic N form is selected as treatment because there are many other organic N forms are available? It is somehow reflected in hypothesis, but a clear statement would help. [Alanine is a simple model organic N compound that is a major component of proteins – the main non-synthetic source of organic N in soils. Alanine has been used previously as a model organic N in studies investigating both plant (Hill et al. 2011, PLoS1) and microbial (Farrell et al. 2011, GBC) organic N uptake, and has been shown to be intermediate in turnover time relative to a number of other amino acids (Farrell et al. 2014, SBB). Urea was used as the alternate organic N compound as it is a commonly used N fertiliser, especially in irrigated cotton system, where its high N density is beneficial for transport logistics. We have added an explanatory sentence detailing this after the hypotheses.]

• You have used labelled N and C to exactly quantify the N uptake by cotton, I suggest making a graphical abstract showing how much proportion of N uptake through several N forms. [completed]

• L123, How the 13C information used in N uptake (equation 1-4)? [Line 129 explains that the equations’ pronumerals should be substituted when calculating carbon. Alternatively, we could list each equation twice next to each other with each of the appropriate pronumerals if that is preferred]

• L188, present r coeff value. Is this a no correlation or non-significant correlation? [There is no correlation is demonstrated in the figure and thus we do not report a value in text]

• L264, L266, Fix syntax error (eg space and spelling) [We believe this is now fixed]

• This is 180 minutes research suggesting a N uptake preference of cotton. Please justify why only 180 minutes, people working on isotope might understand but it is good to provide some rationale to clarify wider audience. [Due to the well documented rapid turnover of amino acids and other low molecular weight dissolved organic N compounds in soil, longer term pulse-chase experiments risk only observing uptake dominated by N (and C) from the target compounds after extracellular mineralisation of the target labelled compound. The very short-term nature of this experiment and the sand matrix growth medium were chosen to best probe N uptake dominated by intact compounds rather than the secondary products of extracellular mineralisation. A short sentence explaining this has been added to the end of the Experimental Design sub-section.]

---

## [Editor Report · Decision Letter 2]

1 Oct 2025

Nitrogen uptake preference of cotton (Gossypium hirsutum L.)

PONE-D-24-59331R2

Dear Dr. Farrell,

We’re pleased to inform you that your manuscript has been judged scientifically suitable for publication and will be formally accepted for publication once it meets all outstanding technical requirements.

Kind regards,

Roshan Babu Ojha

Academic Editor

PLOS ONE

Additional Editor Comments (optional):

Thank you very much for addressing reviewer and editor comments and suggestions.
---

## [Editor Report · Acceptance letter]

PONE-D-24-59331R2

PLOS ONE

Dear Dr. Farrell,

I'm pleased to inform you that your manuscript has been deemed suitable for publication in PLOS ONE. Congratulations! Your manuscript is now being handed over to our production team.

Kind regards,

on behalf of

Dr. Roshan Babu Ojha

Academic Editor

PLOS ONE